# Satisfaction of Online University Education during the COVID-19 Pandemic

**DOI:** 10.3390/healthcare11101421

**Published:** 2023-05-14

**Authors:** Lucía Ortega-Donaire, Jessica Bailén-Expósito, Cristina Álvarez-García, Isabel M. López-Medina, Carmen Álvarez-Nieto, Sebastián Sanz-Martos

**Affiliations:** Department of Nursing, Faculty of Health Sciences, University of Jaen, 23071 Jaén, Spain; lortega@ujaen.es (L.O.-D.);

**Keywords:** students, nursing, e-learning, COVID-19, satisfaction, teaching

## Abstract

The rapid global spread of the coronavirus disease affected the education sector, which had to adapt its teaching methodology to an online scenario to curb the increase in infections. This study aimed to determine the satisfaction level of university students regarding the online education they received during the COVID-19 pandemic. A descriptive cross-sectional study was carried out with nursing degree students. Convenience sampling was used because of the accessibility of the sample, estimating 168 as the minimum sample size needed. The satisfaction level was measured using the “Questionnaire on Satisfaction of University Students with Online Training” (CUSAUF). The sample consisted of 215 participants. The overall level of satisfaction was high. Subject content and teaching methodology were the most highly rated aspects, while aspects related to lecturer-student communication and communication among students were rated the worst. The Cronbach’s α for the satisfaction scale was 0.94, showing high internal consistency and reliability of the sample. Online education could be considered an alternative methodology for teaching learning modules or complementing face-to-face training in future training programs.

## 1. Introduction

The new health situation caused by COVID-19 led to widespread confinement, and face-to-face education had to adapt to a new scenario of online teaching. Thus, 850 million students worldwide at all educational levels changed their teaching methodology [1].

Universities with face-to-face training had to quickly adapt their teaching to a completely online modality, using synchronous communication tools such as videoconferencing and asynchronous tools organized through online training platforms such as Google or Moodle. To date, many universities have offered a portion of their degrees via e-learning with a generally good level of satisfaction, although it is necessary to study the best ways to adapt education to this new reality [2,3].

As expressed by students from different universities, digital literacy has various benefits, such as greater flexibility and versatility in learning and the possibility of participating in training at a preferred time. However, for students to acquire the needed competencies, teachers must have sufficient training and skills related to the management of online teaching [2,4,5,6]. Moreover, issues such as technical and financial support, training, improved working conditions, technological background, skills, copyright protections, and professional development are always important when implementing e-learning in public universities [7].

On the other hand, for an educational program to be effective, in addition to knowledge, other variables must be addressed, such as learning procedures and attitudes. In biomedical sciences, an education based on real cases or simulated examples is also essential to acquire the skills, technical abilities, and knowledge needed to move from theory to reality or practice [2,3,4,7,8,9,10,11].

The meta-analysis of Voutilainen et al. [12] found a higher score in the satisfaction of nursing students using e-learning methods compared to conventional face-to-face class methodology. Even so, despite e-learning being a promising technology that allows the virtual organization of all the materials and deals with topics discussed face-to-face, students lack an active role in the learning process. However, it can be supported with blended learning, defined as the combination of online material with the real experiences of students using case-based learning [5,13]. Nevertheless, some authors have found factors that cause anxiety and dissatisfaction in students with online training methodologies, citing the lack of feedback, absence or ambiguity in the instructions provided by teachers, and little transparency in the evaluation criteria [6].

In the case of healthcare degrees, previous experience in online education is scarce [10]. An essential part of healthcare training involves learning and developing clinical skills and abilities, and the analysis of teaching in these degrees during the COVID-19 pandemic is particularly important [2,3,4,7,10,11]. A study on undergraduate nursing and medical students in Uganda [14] assessed the acceptance and perceived barriers to e-learning during the COVID-19 pandemic. Half of the students considered that e-learning reduced the quality of the knowledge achieved and that this methodology was not efficient. The barriers found included a poor internet connection or a lack of ability to handle digital platforms; however, it was also shown that satisfaction with this methodology was greater the more online skills students had. The awareness and training of students and teachers on e-learning and the use of existing learning platforms are important to improving attitudes and the use of e-learning. A combination of the online methodology and the use of downloadable learning materials will overcome the challenges associated with the variable quality of internet access in the country. Another study carried out at a private university in South Korea [15] with nursing students analyzed studentsself-efficacy after an educational program with e-learning methodology on simulated basic life support, reporting improvements in terms of knowledge and attitude toward a scenario of a cardiorespiratory arrest and early initiation of CPR maneuvers. In this study, an educational strategy was used that included a mixed methodology with integrated videos and face-to-face conferences, thus improving the knowledge and attitudes of nursing students regarding CPR. However, another study that evaluated the long-term effects of e-learning-based CPR education showed that knowledge and self-efficacy decreased significantly after 3 months, suggesting that continued education and training are essential; so, according to the authors of this study, it would be advisable to establish a retraining program to maintain knowledge and self-efficacy for a longer period of time after 3 months [16].

Therefore, as the training of nursing degrees has traditionally required face-to-face training, given the learning needs and development of clinical strategies and skills, the pandemic circumstances that afflict us pose a challenge for adapting the traditional methodology to e-learning. This adaptation involves uncertainty, and there is a scarcity of results presented in the literature. In addition, it is important to determine the satisfaction of the students with the acquisition of the competencies proposed in this degree. Therefore, this study aimed to evaluate the level of satisfaction of university students enrolled in the nursing degree with the online theoretical and practical teaching provided during the COVID-19 pandemic.

## 2. Materials and Methods

### 2.1. Design

A descriptive cross-sectional study was carried out. The study followed the recommendations of the STROBE (Strengthening the Reporting of Observational Studies in Epidemiology) guideline.

### 2.2. Sample

The target population consisted of undergraduate students enrolled in the nursing degree program at the University of Jaen during the 2020/2021 academic year. The inclusion criterion was that the participants should have received at least six months of face-to-face university teaching prior to changing to the online teaching methodology. A convenience sampling was used because of the accessibility of the sample of students in the 2nd and 3rd years of the nursing degree [17,18]. Based on previous research, a sample size calculation was established to detect a difference of 0.5 points in the level of satisfaction, a standard deviation of 0.8 points, a confidence level of 95%, and a minimum power of 80%, with a minimum of 84 participants per group, or 168 in total [6,10,19].

### 2.3. Instrument

The information was collected through an online questionnaire accessed through a Quick Response code during university nursing classes. The questionnaire was composed of four blocks (Appendix A):

Sociodemographic variables: sex, age, and academic year.

Personal environment: living with minors, living with older adults, and family relationships with both.

Variables related to previous online training: have received previous online training and a numerical score from 0 to 10 that they attribute to this training.

Satisfaction level with online training: measured using the Questionnaire of Satisfaction of University Students with Online Training (CUSAUF) validated in Spanish, consisting of 29 items measured on a Likert scale of 1 to 4 points. The scale obtained an internal consistency index measured by Cronbach’s alpha statistic of 0.91 [6]. The scale’s content validity was determined by the judgment of 14 experts in educational technology and new technologies applied to education from different Spanish universities. The original scale was validated with young student teachers from a Spanish university. The age range was 19 to 21 years old. The scale score ranges from between 29 and 116 points. This scale has been used to measure the level of satisfaction with the online training received by Spanish university students, showing high reliability values [10,19].

The questionnaire comprises five theoretical dimensions assessed by the original experts validation that measure the main aspects to be developed in the teaching:General aspects of the subject: the clarity of the subject syllabus and the proposed learning activities are evaluated;Teaching staff: The ability of the teaching staff to deliver the content is assessed, both in terms of their knowledge and attitudes and their ability to adapt to the online environment;Contents: It assesses the content volume, the presentation’s clarity, and the students’ ability to understand it;Online communication: It measures the ease with which communication can be established between teachers and students and among students by creating an appropriate virtual environment;Aspects related to the teaching platform used: The functionality of the training platform used is evaluated [6].

The level of satisfaction was categorized as follows: excellent (≥90% of the maximum score), good (70–89% of the maximum score), and low satisfaction (≤69% of the maximum score).

For information collection, the nursing degree lecturers, who teach in the second, third, and fourth years, were asked to request the time necessary to fill in the scale during their lectures using the Quick Response code generated. During data collection, at least one research team member was present to answer any questions or problems related to the computer application that might arise. The completion of the questionnaire lasted approximately 10 min. Anonymity was guaranteed throughout the collection and processing of the information.

### 2.4. Data Analysis

Descriptive statistics for all data and the questionnaire score were calculated. A bivariate analysis was performed, with the score of the satisfaction with online training scale as the dependent variable. The sociodemographic variables, personal environment variables, and previous experience with online training were set as independent variables. The nonparametric Mann–Whitney U or Kruskal–Wallis H tests were used depending on the number of independent variable categories. All analyses were performed with the SPSS 24.0 statistical program, and a value of *p* < 0.05 was established as the level of statistical significance.

### 2.5. Ethical Considerations

This study was approved by the Institutional Review Board of the University of Jaen (DIC.20/14.PRY). An information sheet was given to the participants. If they were happy to participate, students completed and signed an informed consent form prior to undertaking the session. Students were not obliged to participate and were reassured that this would not affect their progress or success in their course of study. The confidentiality of personal data was guaranteed. Participants were informed and agreed to have their data published in a scientific article.

## 3. Results

### 3.1. Descriptive Analysis

The initial sample consisted of 224 participants in the 2nd, 3rd, and 4th years of the nursing degree at the University of Jaen. The response rate was 88.2%. Of these, 9 participants were eliminated for not fully completing the questionnaire, resulting in a final sample of 215 participants. The mean age was 21.95 years (standard deviation [SD]: 4.32). The age range of the participants was 33 points, with a minimum age of 18 years and a maximum age of 51 years. The majority were female (75.8%), single (97.7%), and 29.8% had received online training prior to that given at the university (Table 1).

The scale showed a Cronbach’s alpha value of 0.942. The total mean score of the satisfaction scale was 88.37 (SD: 14). The highest-rated categories were the general aspects of the course, with a mean rating of 3.17 (SD: 0.57) points; aspects related to the contents, 3.17 (SD: 0.50); and aspects related to the virtual environment, 3.19 (SD: 0.50).

The overall level of student satisfaction with the online training was good, with 80.9% of the total student body reporting an excellent or good level of satisfaction (Table 2).

The items with the lowest level of satisfaction were item 10, “The professor-tutor provided adequate encouragement and stimulated participation,” and item 24, “The virtual spaces for informal communication among classmates were adequate”; both are related to communication and interaction between students and faculty during the sessions. On the other hand, the items related to more vertical teaching where the teacher provides information that is sought to be of quality, such as item 5, “The teacher-tutor of the online modules had a good command of the subject,” and item 6, “When necessary, the tutor-teacher gave information and explained the contents presented”, were the best rated (Table 3).

### 3.2. Bivariate Analysis

The bivariate analysis found that the level of satisfaction was significantly higher in those participants who had taken online training courses before (94.86 ± 13.56 vs. 85.62 ± 13.31; *p* < 0.01) and in those who lived with minors in the same household (92.95 ± 13.77 vs. 86.42 ± 13.69; *p* < 0.01). In addition, there was a statistically significant weak correlation between the total satisfaction scale score and the score of participants exposed to previous online training (Rho = 0.355; *p* = 0.004). However, no statistically significant differences were found according to the academic year or within the group of 4th-year participants who were distributed among 4 elective subjects (*p* = 0.341) (Table 4).

## 4. Discussion

### 4.1. Main Findings of This Work

This study aimed to determine the satisfaction level of nursing degree students at the University of Jaen with the online theoretical and practical education they received due to the COVID-19 pandemic. Overall satisfaction levels were good/excellent, with an average of 3.13 points out of 4; this result is inconsistent with that found in two previous investigations at Spanish universities [20,21], in which mean satisfaction values of 2.34 and 1.43 points out of 5 were obtained. The timing of the survey can explain this difference. The previous studies were conducted during the second semester of the academic year 2019–2020, and the present research was conducted in the academic year 2020–2021, during which there was more information and experience about the use of online teaching methodologies. A key element in the satisfaction and success of online education is the technical ability of the platform and teachers’ online tools, and they were more accustomed to their use and had received more training for the 2020–2021 academic year. Technological literacy is also increasing, making the population feel more comfortable gaining skills using information and communication technologies and new technologies.

Another aspect that may explain the differences found compared to previous studies is the study population. The present study surveyed nursing degree students, while previous studies selected undergraduate students from different university degrees [20,21]. This variability in the sample may lead to a lower level of satisfaction compared to our research because the sample consisted only of nursing students, which may lead them to feel more identified with the health crisis that caused the change in teaching methodology [22].

A previous study at the University of Jaen found an overall satisfaction level of 2.67 points out of 5, whereas if we analyze the data of the Faculty of Health Sciences with nursing and physiotherapy degree students, the overall level of satisfaction was 2.85 out of 5 [23]. Another difference in our research, in comparison with previous studies, is the scale used and the range of scale scores. The previous studies in Spanish universities [20,21,23] used a Likert scale of 1 to 5 points, while we used a maximum value of 4 points. The scale was also different, so different aspects of the student satisfaction domain were evaluated.

Regarding sociodemographic variables such as age, internet use increased in all age groups in 2020 compared to the pre-pandemic period. During the pandemic, internet use among young people after 16 to 24 in 2020 was practically universal (99.8%), with this use decreasing from age 55. Regarding computer skills, the age group between 16 and 24 years old had advanced by 75.7%, with men having them to a greater extent (45.3%) compared to women (43.0%) [24]. The participants in our research were in this age range, which may affect the higher satisfaction level due to the greater use of information and communication technologies in performing regular activities.

In terms of gender, no great inequality has been observed in internet use during the pandemic, with 83.8% of women and 82.4% of men using it daily. Although women outperform men by 1.2 points when using it several times a day. Regarding activity on the internet during 2020, women showed a greater preference for health topics, education, or social, while men were more likely to play or download games, sell goods or services, or use electronic banking, making clear the unequal distribution between genders [24]. Another possible difference, according to Aguilar et al. [25], is that men have greater skills than women in using new technologies, so they need to improve creativity and the innovation of technologies. Moreover, according to Bezak et al. [26], the female population has a greater professional, social, and personal burden than men, which also influences the development of work and the use of information and communication technologies by women.

We found higher satisfaction with online teaching for students living with children at home. The COVID-19 pandemic caused a forced rearrangement of the conditions we found ourselves in, including receiving classes from home. Previous research showed that a gender gap was present before the pandemic, but it was aggravated as a consequence of it, although, on the other hand, the pandemic may have made it easier for people who have to care for children to study [27,28,29].

The main barriers the nursing degree students perceived in this study referred to the loss of interaction with teachers and classmates, as indicated by the items with the lowest scores referring to the communicative aspects. While communication between teachers and students is always important, we would like to highlight that the students have understood that this perception of a lack of interaction was due to unprecedented exceptional conditions and that they should have tried to work more independently [26]. The creation of an interactive learning environment might solve the loss of communication; however, here there is large variability in the ability of teachers to create these virtual environments, as well as the skills or abilities of each student to adapt to the new situation and interact with their peers and teachers through this new environment, which can generate rejection in the initial stages due to a lack of knowledge and the required skills [30,31].

As a result, we faced the need to train students and teachers in the new teaching modality. Students felt isolated when receiving the teaching, and the teaching staff stated that they had difficulties adapting and teaching synchronously and asynchronously, which considerably affected the quality of the teaching received by the students. This can be seen in the satisfaction level with the adaptation of the undergraduate students reported in another study [21], which was 2.39 out of 5 points. In our research, the general adaptation of students and teachers to online teaching was good, with the lowest scoring item in the dimension of satisfaction with the platform related to the platform response time, as the increasing demand experienced by the platforms could produce momentary saturations that required more time to access them. The aspects related to the teaching staff were positively evaluated, emphasizing the effort they made to adapt their classes to the new teaching methodology [23,31,32].

Interaction with teachers was not the only low-rated item; communication among peers also received a low score. During the pandemic, the students stated that they lost closeness and communication with their classmates, despite using online communication tools [19,33]. This loss of interaction with peers may affect them beyond the development of academic competencies. In this sense, the research by Bączek et al. [4] states that no differences were found regarding the student`s perception of the development of theoretical knowledge competencies; however, they found a loss of communication skills with their peers and with patients, as they were unable to interact with them in the clinical care placements [34].

In addition to the loss of students’ communication skills, in healthcare degrees, there is another important gap perceived by students as a result of the change of teaching modality, such as a loss of the development of technical competencies due to a lack of practice [35]. Thus, in the research by Bączek et al. [4], students in the first years show that the main negative aspects of online teaching are the technical aspects of access or management of the platform or those related to communication with teachers and classmates. In contrast, the students in more advanced courses show another barrier, which is more related to the loss of technical skills associated with the practice of care.

Another aspect to highlight is the perception of the students at the Complutense University of Madrid [21]. They perceived an increase in workload due to the modification of the evaluation criteria when adapting them to the new online methodology. Related to this, the modification of the evaluation methodology has been another aspect that has generated the most dissatisfaction among the students. Previous research with dental students [36] reported that evaluations within the teaching seminars were modified to evaluations through teaching platforms that did not allow them to obtain complete learning [37,38]. Participants from developed countries with good internet connections had significantly higher satisfaction scores with online education compared to students from developing countries. These differences in the ability to use online assessment platforms, the quality of the computers from which the assessment is performed, or the internet connection will generate differences in student scores that are not related to the level of mastery of the subject but related to elements external to the assessment [6].

On the other hand, the aspects most highly valued by the students in our research were the mastery of the subject, the concepts explained by the professor, the provision of correct information, the explanation of the contents, and the resolution of all questions. The adaptation of the institutions and the teaching staff to the new teaching methodologies was positive and good, highlighting that the teaching aspects had a high level of satisfaction. This rapid adaptation was one of the main positive aspects highlighted in previous research [6,19]. Several studies show students’ general satisfaction with this new teaching scenario emerged unexpectedly due to COVID-19, where they value the teaching received as positive [39,40]. In our study, the students who had previous experiences with some online education had significantly higher satisfaction levels, demonstrating the rapid adaptation to online teacher training, which matches the Olum et al. report [18]. Although there was a loss of interaction, which was the main perceived barrier, the communication tools and the students’ feelings were positive, due to students perceiving a greater concern on the part of teachers to clarify concepts, the use of alternative tools to overcome the distance barrier, and the use of different teaching methodologies to achieve all the training objectives [41].

### 4.2. The Implications of These Results in the Field of Study of the Research

Online teaching can be an effective alternative for teaching in university education; however, in nursing degrees, due to their specific characteristics, a more in-depth study is needed to break down the barriers related to the lack of interaction with faculty, peers, patients, and the healthcare environment where technical skills can be developed. The improvements in these areas are key to achieving academic achievements among nursing students since the satisfaction level with online education has been found to be a predictor [42]. A well-accomplished course assignment is the most influential facilitating factor for students [43]. Universities should continue to offer guidance and training to teachers to improve the use and suitability of e-learning tools to different degrees [32,39].

Our research has highlighted some important areas to be improved in future online education; they can be summarized in two topics: (1) communication, which includes communication with peers and educators; (2) online platforms, due to technical problems and the user’s inexperience. Along these lines and with implications for practice, we propose aiming to minimize these problems as online education is here to stay and possible solutions could be developed. We propose improving platform chats, and educators have to motivate their students to work in pairs or groups to improve communication. Additionally, starting the course with a mini-training to understand the online platform and decrease the nervousness produced when using something you do not know well would be beneficial. Simulations or virtual reality technologies could be used to enhance the experience and immersion in the subject since previous meta-analyses showed that this type of education is more effective than other education methods in areas of skills, satisfaction, confidence, and performance time [44].

Despite our study’s strengths and identification of areas for improvement in online nursing education, some limitations must be mentioned. The main limitations of our research were its voluntary nature; the participants with a higher level of satisfaction or who were more motivated at the time of the survey could be the ones who filled in the data, which may cause an overestimation of the score on the attitude scale. Secondly, a separate evaluation of satisfaction with the online teaching of theoretical and practical content would be necessary to allow us to assess, in a more in-depth manner, the degree of satisfaction with the development of practical technical skills in health care degrees. Finally, we should be cautious about extrapolating from our research results due to the sample’s non-probabilistic selection.

### 4.3. Future Lines of Research

As future lines of research, it would be advisable to carry out studies with a separate evaluation of the satisfaction with online theoretical and practical, in a qualitative way, thus allowing a deeper assessment of the degree of satisfaction with the development of practical technical skills in health degrees. It is also important to assess satisfaction with the online assessment system and determine which one guarantees the most objective and transparent way of measuring the acquisition of competencies by students through online teaching.

## 5. Conclusions

In conclusion, the degree of satisfaction with online teaching was high, specifically in aspects related to the subject, the contents presented, and the virtual platform. Some areas needing improvement in nursing online education include communication with peers and teachers, the development of better online platforms, and improving the user’s abilities. In addition, more research should be conducted on the effectiveness and viability of alternative teaching methodologies that can be combined with face-to-face or online teaching to overcome these problems and achieve greater motivation, participation, and student learning. Online education will remain with us for a long time, so it is necessary to address these issues and train students and educators in online skills since this variable is correlated with satisfaction in this type of education. In addition, virtual simulations should be included in online clinical nursing skills training, and the effectiveness of training and competence acquisition, as well as student satisfaction and self-efficacy with this type of training, should be investigated.

## Figures and Tables

**Table 1 healthcare-11-01421-t001:** Characteristics of the sample.

Variable	Category	N (%)	M (SD)
Sex	Men	52 (24.2%)	
	Women	163 (75.8%)	
Age (years)			21.95 (4.32)
Academic Year	2nd	100 (46.5%)	
	3rd	68 (31.6%)	
	4th	47 (21.9%)	
Previous online training	Yes	64 (29.8%)	
	No	151 (70.2%)	
Score			7.33 (1.84)
Living with minors	Yes	64 (29.8%)	
	No	151 (70.2%)	
Family relationship	Brother/Sister	56 (27.9%)	
	Son/Daughter	4 (1.9%)	
Living with older adults	Yes	12 (5.6%)	
	No	203 (94.4%)	
Family relationship	Parents	4 (1.9%)	
	Grandparents	8 (3.7%)	

N—frequency; M—mean; SD—standard deviation.

**Table 2 healthcare-11-01421-t002:** Student satisfaction levels regarding online training.

Category	N (%)
Excellent	56 (26%)
Good	118 (54.9%)
Low	41 (19.1%)

**Table 3 healthcare-11-01421-t003:** Satisfaction scale score.

Dimensions	M	SD
Aspects related to the subject	3.17	0.57
Item 1: The program of the course has been adequate.	3.21	0.58
Item 2: The work and practices of the different modules have been valuable in putting into practice the knowledge acquired.	3.13	0.69
Aspects related to the teaching staff	3.09	0.44
Item 3: The teacher-tutor of the online modules made it easier for me to understand the technical issues of the platform.	3.12	0.63
Item 4: I consider the teacher-tutor’s use of the different online resources to be adequate.	3.14	0.63
Item 5: The tutor-teacher of the online modules had a good command of the subject.	3.25	0.64
Item 6: When necessary, the tutor/teacher gave information and explained the contents presented.	3.38	0.55
Item 7: The tutor/teacher gave appropriate feedback on the activities performed.	3.20	0.56
Item 8: I consider the teacher/tutor’s explanation of the rules of operation of the training environment to be adequate.	3.18	0.59
Item 9: The public or private recommendations on the work and the quality of the work by the teacher/tutor were correct.	3.04	0.76
Item 10: The tutor/teacher provided adequate encouragement and stimulated participation.	2.46	1.11
Item 11: Activities were carried out to facilitate knowledge among the students who were part of the online modules.	3.09	0.75
Content-related aspects	3.17	0.5
Item 12: The different contents presented were current.	3.29	0.58
Item 13: The volume of information is sufficient for training in the different contents presented.	3.26	0.64
Item 14: The contents presented have been easy to understand.	3.08	0.66
Item 15: I think the originality of the contents offered was adequate.	3	0.75
Item 16: The relevance of the contents from the theoretical point of view was adequate.	3.15	0.66
Item 17: The relevance of the contents from the practical point of view was adequate.	3.19	0.65
Item 18: I consider the contents to be pleasant.	3.22	0.61
Item 19: The relationship between the objectives and the contents presented was appropriate.	3.22	0.61
Item 20: The relationship between the timing and the contents offered was appropriate.	3.08	0.76
Item 21: I consider the scientific and didactic-educational quality of the contents addressed to be adequate.	3.22	0.59
Aspects related to communication	2.98	0.57
Item 22: Communication with the teachers-tutors has been easy for me through the communication tools: mail, photo, virtual teaching platform.	3.30	0.62
Item 23: I found it easy to communicate online with the rest of my classmates in my environment.	2.97	0.86
Item 24: The virtual spaces for informal communication among classmates have been adequate.	2.67	1.13
Aspects related to the virtual platform	3.19	0.5
Item 25: The technical operation of the virtual environment is easy to understand.	3.24	0.56
Item 26: I consider the online teaching platform adequate because I found it easy to navigate.	3.31	0.66
Item 27: The aesthetic quality of the environment (size, font, colors...) was adequate.	3.31	0.55
Item 28: There is adequacy between the different aesthetic elements of the platform (text, images, graphics).	3.17	0.61
Item 29: The response times of the platform (waiting to access a link, access to the different tools) were adequate.	2.93	0.78
Total score	88.37	14
Mean score	3.13	0.43

M—mean; SD—standard deviation.

**Table 4 healthcare-11-01421-t004:** Bivariate contrast.

Variable	Category	M ± SD	Contrast	*p*-Value
Sex	Men	88.54 ± 14.35	*Z* = −0.131	0.896
	Women	88.31 ± 13.94		
Age			Rho = 0.052	0.451
Academic year	2nd	89.65 ± 10.55	*Χ*^2^ = 3.157	0.206
	3rd	85.51 ± 140.4		
	4th	89.77 ± 19.19		
Previous online training	Yes	94.86 ± 13.56	*Z*= −4.271	<0.001
	No	85.62 ± 13.31		
Score			Rho = 0.332	0.007
Living with minors	Yes	92.95 ± 13.77	*Z*= −3.189	0.001
	No	86.42 ± 13.69		
Family relationship	Brother/Sister	92.93 ± 13.42	*Z* = −0.430	0.667
	Son/Daughter	93.25 ± 21		
Living with aged	Yes	89.08 ±13.66	*Z* = −0.119	0.905
	No	88.33 ±140.6		
Family relationship	Parents	86.75 ±18.48	*Z* = −0.340	0.734
	Grandparents	90.25 ±11.93		

M—mean; SD—standard deviation.

## Data Availability

All data generated or analyzed during this study are included in this published article [Appendix A].

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
