# Peer review of "Satisfaction of Online University Education during the COVID-19 Pandemic"

_healthcare, 2023, doi:10.3390/healthcare11101421_

Round 1
Reviewer 1 Report
Congratulations to the authors for the excellent data analysis and presentation of the results. Their handling of the methodology is excellent and the paper provides a sound analysis.
However, the whole theoretical framework part is a bit disappointing. An article focusing on the perception of nursing students is awaited. That is why it has been submitted to this journal and, moreover, some paragraphs in the article talk about it. However, the first paragraphs are full of platitudes (starting with the first paragraph on COVID-19 which is irrelevant to the article and rather naïve). These paragraphs should be used to provide a review of the literature on the specific problems nursing students face when they are forced to do their studies online. In other words, I propose to remove all the parts that talk about general cases and to deepen the study on the specific aspects that concern nursing students. In my view, the article should start from line 83 onwards. I propose to extend that explanation and, above all, to go deeper into specific aspects, i.e. what kind of educational strategies have been used in those countries and experiences cited to try to compensate for the fact that nursing students did not have face-to-face placements.
Although the study instrument is very general, a simple satisfaction questionnaire, I believe that this orientation towards specific nursing issues could improve the discussion (also very general and citing various works on topics far removed from the focus of the study) and make it more precise on specific issues that affect nursing students.
Often, by trying to adjust our studies to already validated instruments, which are often too general, we lose focus of the questions that are really interesting, novel and that can contribute something relevant to the research community. I believe that this article, even if it uses one of those general instruments, can gain in interest and relevance if, at least, its entire theoretical design is focused on the suggested topics.
Author Response
Reviewer 1 |
||
Comment |
Response |
Page/line number |
The whole theoretical framework part is a bit disappointing. An article focusing on the perception of nursing students is awaited. That is why it has been submitted to this journal and, moreover, some paragraphs in the article talk about it. However, the first paragraphs are full of platitudes (starting with the first paragraph on COVID-19 which is irrelevant to the article and rather naïve). These paragraphs should be used to provide a review of the literature on the specific problems nursing students face when they are forced to do their studies online. In other words, I propose to remove all the parts that talk about general cases and to deepen the study on the specific aspects that concern nursing students. In my view, the article should start from line 83 onwards. I propose to extend that explanation and, above all, to go deeper into specific aspects, i.e. what kind of educational strategies have been used in those countries and experiences cited to try to compensate for the fact that nursing students did not have face-to-face placements. |
Thanks for the contribution. The entire theoretical framework section has been modified, eliminating irrelevant information, contributing new data to the problems that nursing students face when they are subjected to this new online modality. More emphasis has been placed on the perception of nursing students. |
Page 1-3 |
Although the study instrument is very general, a simple satisfaction questionnaire, I believe that this orientation towards specific nursing issues could improve the discussion (also very general and citing various works on topics far removed from the focus of the study) and make it more precise on specific issues that affect nursing students. |
The discussion has been modified to focus more on specific aspects of online nursing education, barriers and solutions. Information that is too general has been removed. |
Pages 8-11 |

Reviewer 2 Report
The submitted manuscript is interesting, however, it presents a series of aspects that could be improved, which I list below, because in the current form in which it is presented, from my point of view, it is not publishable:
SUMMARY:
- Irrelevant information appears, such as the allusion to Cronbach's α. - However, other information that should appear does not appear, such as the sample size... - I recommend redoing the summary following the introduction, objectives, method, results and conclusions sections.
INTRODUCTION: - Review throughout the entire manuscript that the references comply with the journal's standards. - The introduction should end with the research questions or hypotheses that are intended to be achieved with this work. - It is not clear why do this research
METHOD: - This is the weakest section of your manuscript - When the participants are described, it would be convenient to add a table showing the main socio-demographic characteristics of the sample. - I recommend structuring this section in: or Design or sample or Instruments o Data analysis
DISCUSSION AND CONCLUSIONS: - I recommend adding the discussion in the following sections: o Main findings of this work o The implications of these results in the field of study of the research o Future lines of research
Author Response
Reviewer 2 |
||
SUMMARY: Irrelevant information appears, such as the allusion to Cronbach's α. - However, other information that should appear does not appear, such as the sample size... - I recommend redoing the summary following the introduction, objectives, method, results and conclusions sections. |
The abstract has been modified, but we have maintained the structure in a single paragraph as set out in the journal's guidelines. |
|
INTRODUCTION: - Review throughout the entire manuscript that the references comply with the journal's standards. - The introduction should end with the research questions or hypotheses that are intended to be achieved with this work. - It is not clear why do this research |
The last paragraph has been modified, making the research question clear. References have been checked. |
Page 3 |
METHOD: - This is the weakest section of your manuscript - When the participants are described, it would be convenient to add a table showing the main socio-demographic characteristics of the sample. - I recommend structuring this section in: or Design or sample or Instruments o Data analysis |
The characteristics of the sample are shown in Table 1, in the results section. The methodology section has been restructured |
|
DISCUSSION AND CONCLUSIONS: - I recommend adding the discussion in the following sections: o Main findings of this work o The implications of these results in the field of study of the research o Future lines of research |
The discussion have been structured in the recommend sections. |
Pages 8-11 |

Reviewer 3 Report
troduction
- The aim needs to be consistent with your topic. Your topic stated, "Satisfaction of online university education during the covid-19 pandemic"; In contrast, the aim stated in the abstract said, " This study aimed to determine the satisfaction level of university students regarding the online training 11 received during the COVID-19 pandemic". You need to determine if it aimed to assess the satisfaction level of university students regarding the online training only or for education in general, including the theory part.
- The abstract needs to include the sample size and setting of your study.
- The instrument of your study needs to be explained further as its validity and reliability.
Methods:
- The sampling of your study and response rate need to mention in the method section.
- The questionnaire's construct and reliability, and validity must be explained thoroughly.
Overall comments
- Please follow the guidelines of the journal, especially in the reference list
Author Response
Reviewer 3 |
||
Introduction: The aim needs to be consistent with your topic. Your topic stated, "Satisfaction of online university education during the covid-19 pandemic"; In contrast, the aim stated in the abstract said, " This study aimed to determine the satisfaction level of university students regarding the online training 11 received during the COVID-19 pandemic". You need to determine if it aimed to assess the satisfaction level of university students regarding the online training only or for education in general, including the theory part. |
The proposed objective has been modified, making it clear that it refers to theoretical and practical education. |
Page 3 |
The abstract needs to include the sample size and setting of your study. |
Added |
|
The instrument of your study needs to be explained further as its validity and reliability. |
Made |
Page 3 |
The questionnaire's construct and reliability, and validity must be explained thoroughly. |
Made. If you need more details about the validation process we can provide them. |
Page 3 |
Methods:The sampling of your study and response rate need to mention in the method section. |
Sampling method was explained at the methodology section. Response rate was added at the results section |
Page 3 |
Overall comments: Please follow the guidelines of the journal, especially in the reference list |
Checked |
|

Reviewer 4 Report
You have written a concise yet thorough paper highlighting your findings and offering some key findings post-secondary educational institutions will need to address if online education is to continue in the Nursing discipline. Your findings flow cleanly from your research methods. This is a study of interest to academics in your field, but also the wider open learning community.
Author Response
Thank you very much

Round 2
Reviewer 2 Report
CONGRATULATIONS